# YOLOv8++ with Weights Pruning for Road Object Detection in Rainy Environment

Anonymous Full Paper
Submission 35

## Abstract

Object detection on roadways is crucial for autonomous driving and advanced driver assistance systems. However, adverse weather conditions, particularly rain, significantly degrade the performance of these systems. This paper presents a novel approach to enhance road object detection in rainy weather scenarios by applying a modified YOLOv8 model. The proposed YOLOv8++ model includes specialized data augmentation techniques to simulate rainy conditions, adjustments in the network architecture to improve robustness against rain-induced noise, and optimized training strategies to enhance model performance. The study leverages BDD100K, Cityscapes and DAWN-Rainy datasets consisting of various road scenarios under different intensities of rain. We systematically augment these datasets to ensure the model learns to identify objects obscured by rain streaks and reflections. Our YOLOv8++ model introduces enhancements in the feature extraction layers, enabling better handling of occlusions and reduced visibility. Extensive experiments demonstrate that our model outperforms the baseline YOLOv8 and other state-of-the-art object detection models in terms of mean Average Precision (mAP) under rainy conditions. Additionally, to ensure the model's efficiency and suitability for real-time applications, we apply a network pruning technique, which reduces the model size and computational requirements without sacrificing performance. This research contributes to the field of autonomous driving by providing a more reliable object detection system for adverse weather conditions, enhancing overall road safety.

## 1 Introduction

Road object detection is a cornerstone of autonomous driving systems and advanced driver assistance systems (ADAS) [1]. This technology plays a crucial role in identifying and classifying objects such as vehicles, pedestrians, traffic signs, and obstacles within the driving environment. The accuracy and reliability of these detection systems are paramount for ensuring safety and enhancing the overall driving experience. With the rapid advancements in deep learning, the YOLO (You Only Look Once) family of models [2] has emerged as a leading approach in object detection, thanks to its high speed and precision. YOLO models are renowned for their efficiency as they perform object detection in a single forward pass, unlike two-stage detectors such as Faster R-CNN, which involves separate stages for region proposal and object classification. YOLOv11, the latest iteration in the YOLO series, has further improved upon previous models with enhanced architecture and training techniques, setting new benchmarks for performance.

Despite these advancements, road object detection remains a challenging task, especially under adverse weather conditions like rain [3]. Rain presents unique difficulties that can significantly impair the effectiveness of detection systems. The presence of rain can obscure visibility through the camera lens, creating blurred images that make it harder for detection algorithms to identify objects accurately. Additionally, reflections and glare from wet surfaces can introduce noise and distortions, further complicating the detection process. These issues are compounded by the dynamic nature of rainy weather, where the intensity of rainfall, splashes, and mist can vary widely, making it difficult to maintain consistent performance across different conditions. Moreover, the availability of annotated datasets specifically for rainy weather is limited, which hampers the ability to train and evaluate models effectively for such scenarios.

To address these challenges, researchers have explored various methods to improve object detection in adverse weather. Data augmentation techniques [3] have been employed to simulate rainy conditions and enhance the diversity of training datasets. Specialized network architectures have been designed to improve robustness to noise and distortions. Additionally, the incorporation of sensor data from sources like LiDAR and radar has been investigated to complement camera-based detection. Despite these efforts, there is still a need for more robust and efficient solutions that can effectively handle the complexities introduced by rainy weather.

In this study, we propose a modified YOLOv8 model specifically designed to improve road object detection in rainy weather scenarios. Our approach involves several key modifications aimed at enhancing the model's performance under challenging conditions. First, we employ advanced data augmentation techniques to create a diverse set of training

samples that mimic various rainy conditions. This includes simulating rain streaks, droplets, fog, and glare, which helps the model learn to recognize road objects despite these distortions. By training the model on such augmented data, we aim to improve its ability to detect objects in real-world rainy scenarios. Second, we introduce enhancements to the YOLOv8 [4] architecture to improve feature extraction and robustness. Our modifications include the integration of specialized layers and modules that are designed to handle noise and distortions more effectively. These enhancements aim to improve the model's ability to extract meaningful features from the input images, even when they are obscured by rain or other adverse conditions. By strengthening the feature extraction capabilities, we hope to achieve more accurate and reliable detections.

Furthermore, to ensure that our modified YOLOv8 [4] model is accurate and efficient, we apply a network pruning technique. Network pruning [5] involves removing redundant and less significant parameters from the neural network, resulting in a smaller model size and reduced computational complexity. This process helps to achieve faster inference times, which is crucial for real-time applications in autonomous driving systems. By reducing the number of computations required, pruning enables the model to operate more efficiently on resource-constrained devices, such as in-vehicle computers and embedded systems.

The application of network pruning provides several benefits. Firstly, it leads to a smaller model size, which is easier to deploy and manage in practical systems. A smaller model also consumes less memory, making it suitable for deployment on devices with limited storage capacity. Secondly, faster inference times are achieved through pruning, which is critical for real-time decision-making in autonomous vehicles. Real-time performance is essential for ensuring timely responses to dynamic driving situations, and pruning helps to meet this requirement by reducing the time needed for model predictions. Lastly, network pruning contributes to lower power consumption, which is beneficial for battery-powered devices and overall system sustainability.

The significance of this work lies in its contribution to improving road object detection under rainy weather conditions. By addressing the specific challenges associated with rain, our modified YOLOv8 model enhances the reliability and robustness of object detection systems. This improvement has important implications for the safety and effectiveness of autonomous driving systems, as it ensures more accurate detection of objects even in adverse weather. Additionally, the application of network pruning not only enhances the efficiency of the model but also makes it practical for real-world deployment. Moreover, the techniques and modifications proposed in this study can be extended to other challenging weather conditions, such as snow, fog, and low-light environments. This broadens the applicability of our approach and provides a foundation for future research in developing robust detection systems for various adverse scenarios. The use of advanced data augmentation techniques also contributes to the creation of more diverse and comprehensive training datasets, benefiting the broader research community by providing better resources for training and evaluating models.

# 2 Proposed Model

Modifying YOLOv8 [4] based on compound scaling involves optimizing the architecture to improve performance by adjusting key parameters such as depth, width, and channels. Compound scaling, introduced in models like EfficientNet [6], allows for a systematic way to scale different dimensions of the network simultaneously, leading to a more balanced and effective model. The core idea is to achieve a better trade-off between accuracy and efficiency by uniformly scaling these three aspects rather than scaling them independently. The scaling coefficients are determined through a compound coefficient, which is used to guide how much each dimension should be scaled.

YOLOv8 is already a powerful object detection model, but incorporating compound scaling can enhance its performance, especially for specialized tasks such as road object detection in adverse weather conditions. The modification involves adjusting three key parameters, as listed in Table 1:

- Depth Scaling: This involves increasing or decreasing the number of layers in the network. In YOLOv8, increasing the depth means adding more convolutional layers or residual blocks, which can help the model learn more complex features. However, this also increases computational complexity, so it's essential to find a balance that maintains real-time performance.

- Width Scaling: Width scaling adjusts the number of channels in each layer. By increasing the width, the model can capture more features at each layer, improving its ability to detect smaller or more complex objects. However, increasing the width also increases the memory footprint and computational cost. For YOLOv8, careful tuning of the width parameter can lead to better detection accuracy without significantly compromising speed.

- Max Channels: Max channels refer to the upper limit on the number of channels in any network layer. By optimizing this parameter, the model can be tailored to handle specific tasks more

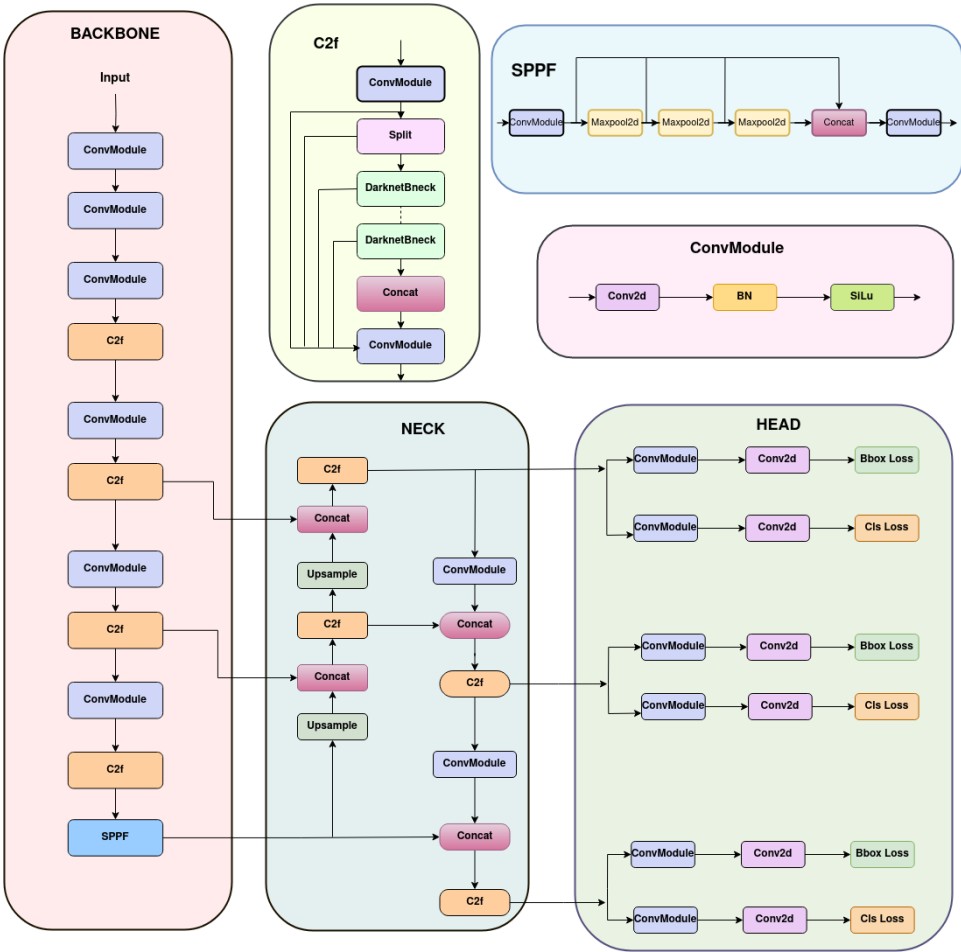

**Figure 1.** Architecture of YOLOv8++

efficiently. For instance, in scenarios like rainy weather object detection, where reflections and low contrast are issues, adjusting the maximum number of channels can help the model focus on the most relevant features without being overwhelmed by noise.

**Table 1.** Compound Scaling Parameters

| Model | Depth | Width | Max Channels |
|---|---|---|---|
| YOLOv8 | 1.00 | 1.00 | 512 |
| YOLOv8++ | 1.25 | 0.80 | 768 |

One ConvModule is added to the existing YOLOv8 architecture. Adding this to the existing architecture aims to reduce the effect of rain streaks, noise and distortion created during the data augmentation method. This module would increase the number of architecture parameters, which are further reduced during the pruning phase. The modified architecture of YOLOv8, which is now YOLOv8++, is shown in Fig. 1. The number of parameters of YOLOv8++ is higher than YOLOv8. Further, the model shifts its focus to reducing the number of pa-

rameters, using weight pruning, without capitalizing much on the accuracy.

Magnitude-based weight pruning is a technique that effectively reduces the size and complexity of neural networks by selectively removing weights with the smallest magnitudes, which are often deemed less critical for the network's performance [7]. When applying this pruning method to the YOLOv8++ model, it enhances computational efficiency without significantly affecting detection accuracy, making it highly suitable for real-time applications, especially in environments with limited computational resources. A detailed description of weight pruning is shown in Algorithm 1.

Applying the suggested changes to form YOLOv8++, which may involve adjustments such as increased depth or width through compound scaling and pruning, can be particularly beneficial. The increased model size from these modifications typically results in a higher number of parameters, many of which may be redundant or contribute minimally to the network's overall performance. By removing these insignificant weights, the pruning process reduces the computational load, leading to a more compact model with faster inference times.

However, it is important to monitor the trade-off between the pruning level and the model's accuracy. If the pruning threshold is set too high, resulting in an excessive number of weights being pruned, the network's accuracy may degrade, especially in complex tasks like detecting road objects under challenging conditions such as rain. This loss in accuracy can be mitigated by fine-tuning the network after pruning, where the remaining weights are adjusted to compensate for the pruned parameters.

# 3 Experimental Evaluation

## 3.1 Dataset Description

For this study, we utilized the BDD100K [8], Cityscapes [9] and DAWN-Rainy [10] datasets, which contain 100K, 5K and 200 annotated images, respectively, depicting a variety of road scenarios, including urban, rural, and highway driving, with diverse lighting and weather conditions. The dataset includes labels for multiple object categories, such as vehicles, pedestrians, traffic lights, and traffic signs, making it suitable for training object detection models in complex environments. BDD100K and Cityscapes datasets are clear weather datasets, while DAWN-Rainy is the real rain dataset.

Synthetic rain generation typically involves overlaying rain streaks, droplets, and splashes onto images while simulating real-world effects like motion blur, light scattering, and refraction. These rain patterns can be generated using various methods, such as procedural rendering, physics-based models, or even generative adversarial networks (GANs). The aim is to replicate the visual distortions caused by rain, allowing models to learn how to identify objects and road elements even under difficult weather conditions.

Augmenting datasets like BDD100K or Cityscapes with synthetic rain (to become BDD100K-Rainy and Cityscapes-Rainy) can simulate diverse rainy conditions (light drizzle, heavy downpours) without needing extensive real-world data collection. This helps train more resilient models to generalize better across different weather scenarios. Such augmented data ensures autonomous vehicles' safe and reliable operation in real-world driving conditions, particularly in areas prone to rain.

To simulate rainy weather conditions and enhance the model's ability to detect objects under adverse weather, we applied the data augmentation shown in Algorithm 2. This augmentation process ensured the model could generalize well to real-world rainy conditions and is applied to the original BDD100K and Cityscapes datasets. The qualitative evaluation of YOLOv8++ over a few sample synthetic rain-generated images is shown in Fig. 2.

---

**Algorithm 1** Weight Pruning Algorithm for YOLOv8++ Model

**Input:**
1: Pre-trained YOLOv8++ model weights $W$.
2: Pruning ratio $r$, the fraction of weights to prune.
3: Pruning criterion: L1-norm or magnitude-based criterion.
4: Fine-tuning dataset $D$, for retraining after pruning.
5: Maximum pruning iterations $K$.
6: Penalty parameter $\rho$ (for regularization-based pruning, if needed).

**Output:** Pruned and fine-tuned YOLOv8++ model weights $W^*$.

---

1: **Initialization:** Formulate the pruning objective as follows:

$$\min_{W} \mathcal{L}(W) \quad \text{subject to} \quad \|W\|_0 \leq k$$

where $W$ is the weight matrix, and $k$ is the target number of remaining weights determined by the pruning ratio $r$.

2: **Compute Weight Importance:** Use the L1-norm or magnitude criterion to calculate the importance of each weight $w_i$:

$$\text{Importance}(w_i) = |w_i|$$

For structured pruning (e.g., filter pruning), compute the importance of each filter $F_j$ as:

$$\text{Importance}(F_j) = \sum_{i=1}^{n_j} |w_{ij}|$$

where $n_j$ is the number of weights in filter $F_j$.

3: **Apply Pruning:** Prune the lowest $r\%$ of weights based on importance scores by creating a binary mask $M$:

$$M[i] = \begin{cases} 0, & \text{if } |w_i| < \text{threshold} \\ 1, & \text{otherwise} \end{cases}$$

Then update the weight matrix:

$$W_{\text{pruned}} = W \odot M$$

where $\odot$ denotes element-wise multiplication between $W$ and the mask $M$.

4: **Fine-Tuning the Pruned Model:** Fine-tune the pruned model $W_{\text{pruned}}$ using the dataset $D$:

$$W^* = \arg\min_{W} \mathcal{L}(W_{\text{pruned}})$$

This helps recover accuracy after pruning.

5: **Evaluate the Pruned Model:** After fine-tuning, evaluate the pruned model $W^*$ to ensure that it maintains high performance in object detection tasks.

**Algorithm 2** Synthetic Rain Generation and Augmentation for Dataset Images

---

**Input:**

1: $X$: Original dataset of images (e.g., BDD100K or Cityscapes)
2: $N$: Number of rain streaks
3: $\theta$: Rain streak angle (in degrees)
4: $L$: Rain streak length (in pixels)
5: $P_{hflip}$: Probability of horizontal flip
6: $S$: Scaling factor range for resizing
7: $\alpha$: Brightness adjustment factor
8: $\sigma$: Standard deviation for Gaussian blur
9: $\beta$: Rain opacity factor

**Output:** Augmented dataset of images $X_{aug}$ with synthetic rain and other augmentations.

---

1: **Step 1: Add Rain Streaks**
2: **for** each image $x \in X$ **do**
3:     Get image dimensions $W, H$.
4:     **for** $i = 1$ to $N$ **do**     ▷ Generate rain streaks
5:         Randomly select a starting point $(x_0, y_0)$, where $x_0 \in [0, W]$ and $y_0 \in [0, H]$.
6:         Compute endpoint $(x_1, y_1)$:

$$x_1 = x_0 + L \cdot \cos(\theta)$$

$$y_1 = y_0 + L \cdot \sin(\theta)$$

7:         Draw a line between $(x_0, y_0)$ and $(x_1, y_1)$.
8:     **end for**
9:     Apply motion blur to rain streaks with Gaussian kernel $G(x, y)$:

$$G(x, y) = \frac{1}{\sqrt{2\pi}\sigma} \exp\left(-\frac{x^2 + y^2}{2\sigma^2}\right)$$

10:     Blend rain streaks with the image using opacity factor $\beta$:

$$I_{aug}(x, y) = (1 - \beta) \cdot I(x, y) + \beta \cdot R(x, y)$$

11: **end for**
12: **Step 2: Data Augmentation**
13: Perform random horizontal flip with probability $P_{hflip}$:

$$I_{flip}(x, y) = I(W - x, y)$$

14: Randomly scale the image by a factor $s \in S$.
15: Adjust brightness with factor $\alpha$:

$$I_{bright}(x, y) = \alpha \cdot I(x, y)$$

16: Apply Gaussian blur to simulate light scattering with kernel size $k$:

$$I_{blur}(x, y) = \sum_{i=-k}^{k} \sum_{j=-k}^{k} I(x + i, y + j) \cdot G(i, j)$$

17: **Step 3: Final Augmented Dataset**
18: Save augmented image $I_{aug}$ in $X_{aug}$.
19: Repeat for all images in the dataset $X$.

---

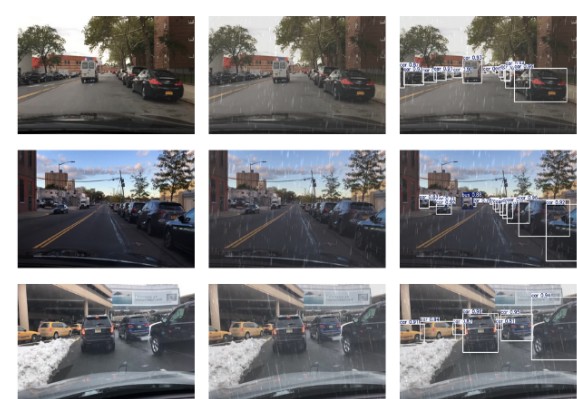

**Figure 2.** Qualitative evaluation of YOLOv8++ over synthetic rain generated and augmented data. Column 1: Original images, Column 2: Rain-augmented images, and Column 3: Bounding boxes generated by YOLOv8++ model

## 3.2  Training and Metrics

The training of the YOLOv8++ model was conducted using the three original (two without rain and one with rain) and two rain-augmented datasets with an 80:20 training-testing split ratio. We employed the Adam optimizer with an initial learning rate of 0.001, which decayed by a factor of 0.1 after every 20 epochs. The batch size was set to 16, and training was performed for 250 epochs. The model was trained on NVIDIA RTX 4090 dual GPUs of 24 GB each, leveraging the mixed precision training to speed up the process while maintaining computational efficiency.

To evaluate the model's performance, we used mAP, the mean Average Precision calculated at the Intersection over Union thresholds of 0.5, to assess the precision and recall trade-off, the number of model parameters and the compression ratio.

## 3.3  Performance Results

Our YOLOv8++ model is compared against the baseline YOLOv8 and other state-of-the-art object detection models over mAP@50, as shown in Table 2. The best and second-best results are marked in bold and underlined, respectively. It can be seen that the proposed YOLOv8++ outperforms YOLOv8 and the recent versions like YOLO-NAS, YOLOv10 and the latest YOLOv11 models. However, when compared in terms of the number of parameters, YOLOv8++ is on the higher side.

The weights pruning-based ablation study is done on the YOLOv8++ model, and the results obtained over mAP@50 and the number of parameters can be seen in Table 3, where the best and second-best results are marked in bold and underlined, respectively. It can be seen that pruning done at 50% has comparable performance with the original YOLOv8++

340 model. The performance of YOLO8++ (mAP@50
341 with 50% pruning - 57.8) is marginally better than
342 YOLOv8 (mAP@50 - 57.6) but with a reduced num-
343 ber of parameters (YOLOv8++ with 50% pruning
344 - 21.897 M params vs YOLOv8 - 43.69 M params)
345 achieving the 2x compression ratio. This makes
346 the proposed model an excellent candidate for real-
347 time object detection in adverse weather conditions,
348 specifically for autonomous vehicles and ADAS.

**Table 2.** mAP@50 results and Params of YOLO variants

| Dataset/Model | YOLO-NAS | YOLOv8 | YOLOv10 | YOLOv11 | YOLOv8++ |
|---|---|---|---|---|---|
| BDD100K | 47.5 | 56.7 | 57.8 | 57.7 | 57.7 |
| Cityscapes | 46.4 | 55.7 | 49.3 | 49.3 | 55.9 |
| DAWN-Rainy | 52.7 | 69.9 | 67.8 | 61.3 | 70.7 |
| BDD100K-Rainy | 46.5 | 57.1 | 57.4 | 57.2 | 57.3 |
| Cityscapes-Rainy | 45.6 | 48.8 | 48.7 | 49.3 | 49.3 |
| Average mAP@50 | 47.7 | _57.6_ | 56.2 | 55.0 | **58.2** |
| Params (M) | 66.90 | 43.69 | _25.89_ | **25.30** | 43.692 |

**Table 3.** mAP@50 results, Params and Compression ratio based on weights pruning

| Dataset/Pruning | 0% | 10% | 20% | 30% | 40% | 50% | 60% |
|---|---|---|---|---|---|---|---|
| BDD100K | 57.73 | 57.71 | 57.70 | 57.71 | 57.74 | 57.46 | 56.89 |
| Cityscapes | 55.94 | 54.96 | 54.75 | 54.32 | 54.59 | 52.72 | 45.83 |
| DAWN-Rainy | 70.72 | 70.48 | 60.80 | 70.67 | 60.88 | 72.43 | 57.80 |
| BDD100K-Rainy | 57.25 | 57.05 | 57.17 | 57.30 | 57.30 | 56.97 | 56.34 |
| Cityscapes-Rainy | 49.32 | 48.63 | 48.62 | 48.65 | 48.69 | 49.31 | 49.15 |
| Average mAP@50 | **58.19** | 57.77 | 55.81 | 57.73 | 56.07 | _57.80_ | 53.20 |
| Params (M) | 43.692 | 39.333 | 34.974 | 30.615 | 26.256 | _21.897_ | **17.538** |
| Compression Ratio | 1x | 1.11x | 1.25x | 1.43x | 1.66x | _2x_ | **2.49x** |

## 4    Conclusion

350 This study presents a significant advancement in
351 road object detection under rainy weather scenar-
352 ios by proposing the YOLOv8++ model with net-
353 work pruning techniques. The combination of en-
354 hanced accuracy, robustness, and efficiency makes
355 our approach a valuable contribution to developing
356 reliable and practical autonomous driving systems.
357 Through rigorous evaluation and comparative anal-
358 ysis, we demonstrate the effectiveness of our modi-
359 fications and provide a solid foundation for future
360 research in this area. Our work addresses critical
361 challenges in adverse weather conditions and paves
362 the way for more reliable and efficient object detec-
363 tion in autonomous driving applications. Magnitude-
364 based weight pruning, when applied to a modified
365 YOLOv8++ model, results in a more efficient net-
366 work by eliminating less important weights. This
367 leads to a leaner, faster model with a reduced com-
368 putational footprint, making it ideal for deployment
369 in real-time systems where resource constraints are
370 critical. The careful balance between pruning and
371 fine-tuning ensures that the network maintains high
372 accuracy while operating more efficiently, particu-
373 larly in demanding environments like autonomous
374 driving under adverse weather conditions.

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
