# OpenReview forum: "YOLOv8++ with Weights Pruning for Road Object Detection in Rainy Environment"
_NLDL.org/2025/Conference — Submitted to NLDL 2025_

### Official Review · Reviewer_KSSN · 2024-09-23
**Lack of novelty and detail in augmentation and architectural adjustments**

**Confidence:** 4

**Summary:**

The paper proposes YOLOv8++, a modified version of YOLOv8 aimed at improving road object detection in rainy conditions. It claims to introduce new data augmentation techniques, make architectural adjustments for handling rain-induced noise, and apply weight pruning to optimize the model for real-time applications. The model is tested using several datasets, including BDD100K and DAWN-Rainy, and reportedly outperforms the baseline YOLOv8 and other detection models in terms of mean Average Precision (mAP) under rainy conditions.

**Strengths:**

Relevant problem: The paper addresses an important issue in autonomous driving: improving object detection in bad weather conditions.

Efficiency considerations: The use of weight pruning is a good approach for improving the model’s real-time performance, which is key for practical applications like autonomous driving.

**Weaknesses:**

Limited detail in augmentation techniques: The paper mentions using advanced data augmentation for rainy conditions but does not give details beyond basic transformations like flipping, color distortion, and blurring. This lack of specific techniques weakens the contribution, especially when other works have explored more rain-specific augmentations.

Unclear architectural modifications: The claimed architectural changes are vague. Simply tuning hyperparameters like depth and width does not qualify as meaningful model innovation. Figure 1, intended to explain the improved architecture, does not highlight what specifically sets YOLOv8++ apart from YOLOv8. Additionally, there is no clear explanation of how these hyperparameters were optimized in practice.

Missing comparative analysis: The paper does not compare its performance with other models specifically designed for rain, even though it acknowledges the existence of such work.

Over-explanation of basic methods: The section on weight pruning is overly long and focuses on basic concepts, offering little new insight. The pruning results do not add much scientific value, as they fail to demonstrate notable improvements or trade-offs beyond what is already known or expected.

Limited scientific contribution: The paper's overall contribution is weak. The augmentation techniques and architectural changes are either too simple or poorly explained, while the pruning results are not particularly innovative or useful.

**Justification:**

While the paper addresses a relevant and practical problem—improving object detection in rainy conditions—it lacks sufficient depth and novelty to warrant acceptance. The augmentation techniques are not clearly explained beyond basic transformations, and the architectural modifications appear to be limited to simple hyperparameter tuning. Moreover, the pruning approach is standard and does not offer new insights or significant improvements. Overall, the paper's contributions are too incremental, with insufficient detail and innovation, making it hard to consider it a substantial addition to the field.

---

> ### Author Rebuttal · Authors · 2024-10-24
>
> Comment 1 -  Limited detail in augmentation techniques: The paper mentions using advanced data augmentation for rainy conditions but does not give details beyond basic transformations like flipping, color distortion, and blurring. This lack of specific techniques weakens the contribution, especially when other works have explored more rain-specific augmentations.
>
> Response -  A detailed algorithm for data augmentation is discussed in Algorithm 2.
>
>
> Comment 2 - Unclear architectural modifications: The claimed architectural changes are vague. Simply tuning hyperparameters like depth and width does not qualify as meaningful model innovation. Figure 1, intended to explain the improved architecture, does not highlight what specifically sets YOLOv8++ apart from YOLOv8. Additionally, there is no clear explanation of how these hyperparameters were optimized in practice.
>
> Response - In Fig. 1, the first ConvModule is added to the existing YOLOv8 architecture. Adding this to the existing architecture aims to reduce the effect of rain streaks, noise, and distortion created by the data augmentation method. This module would increase the number of architecture parameters, which would be further reduced during the pruning phase. Besides, the compound scaling parameters are listed in Table 1 of the revised manuscript.
>
>
> Comment 3 - Missing comparative analysis: The paper does not compare its performance with other models specifically designed for rain, even though it acknowledges the existence of such work.
>
> Response - There are two approaches for tackling Road Object Detection (ROD) in Rainy environments: 1) Remove rain from the image and then apply ROD models, and 2) ROD models are trained on rainy data for detection.
> We have followed approach 2 in this paper. Approach 1 will increase the computational complexity of the model since it involves 2 phases: Deraining the image first and then applying ROD. In this paper, we aim to find a real-time solution for ROD. Comparing approach 1 may not be beneficial as the number of parameters will increase significantly.
>
>
> Comment 4 - Over-explanation of basic methods: The section on weight pruning is overly long and focuses on basic concepts, offering little new insight. The pruning results do not add much scientific value, as they fail to demonstrate notable improvements or trade-offs beyond what is already known or expected.
>
> Response - The weight pruning is now summarized in Algorithm 1.
>
>
> Comment 5 - Limited scientific contribution: The paper's overall contribution is weak. The augmentation techniques and architectural changes are either too simple or poorly explained, while the pruning results are not particularly innovative or useful.
>
> Response -  In the revised manuscript, a detailed algorithm for weight pruning and data augmentation are discussed in Algorithms 1 and 2, respectively. Besides, the architectural changes are clearly explained in the revised manuscript.

---

### Official Review · Reviewer_vnBQ · 2024-10-06
**A very good idea and topic, but unfortunately the work is too lacking**

**Confidence:** 4

**Summary:**

The paper proposes a modified version of the YOLOv8 model, called YOLOv8++, which is designed to improve object detection under adverse weather conditions, and specifically rain.
The proposed modification can be categorised in 3 groups: (1) Integration of specialised layer an modules designed to handle noise and distortion more effectively; (2) Advanced data augmentation techniques to simulate rain streaks, droplets, fog, and glare; and (3) weight pruning to ease complexity and reduce inference time.
All these proposed modification are perfectly sound, and have the potential to substantially improve performance; however, from the paper it is not clear whether they have actually been applied.
There is no mention, other than in the introduction, of any specialised layer or module being added. If such layers/modules are present, their properties are not described anywhere, nor there is any explanation as to why or how they would be specialised to handle noise and distortion, as claimed.
The claim of "advanced data augmentation simulating rain streaks, droplets, fog, and glare" put in the introduction seems also to be unsatisfied, as in the Datasets Description section actually states "we applied basic data augmentation techniques [such as] horizontal flips, scaling, brightness adjustment, and blurring". The authors claim these modification can mimic the effect of rain, which is clearly incorrect.
The weight pruning is indeed applied correctly, but it's clear from Table 2 of the results, that applying pruning reduces the performance of the proposed model back to being equal to the original model without modifications, de facto nullifying any possible improvement. Even without pruning, the improvements of the proposed model shown in Table 1 seem very minor, being restricted mostly to the third decimal of an already very low precision of 50% (toss of a coin). I actually question whether these difference in the average precision are statistically relevant, but no statistical test is presented.
The topic of object detection is definitely and important area of research, especially in the context of driving in adverse weather conditions, and the ideas the authors' propose in the points (1)--(3) above are definitely sound, but their application in the paper is either incorrect, not explained, or appear to not have been done. The results are too weak, and statistical significance tests are not performed, so it is unclear whether there is any actual improvement brought by the proposed method.

**Strengths:**

The paper touches upon the important and actual topic of object recognition in driving under adverse weather conditions. This is a safety-critical application for which additional research is necessary and essential to ensure robust and safe deployment of ML models, either to fully-autonomous driving vehicles or just to assisted driving.
In the introduction, the authors describe their proposed model in terms of three modification to the YOLOv8 model:
(1) Integration of specialised layer an modules designed to handle noise and distortion more effectively;
(2) Advanced data augmentation techniques to simulate rain streaks, droplets, fog, and glare; and
(3) weight pruning to ease complexity and reduce inference time.
These ideas as presented in the introduction gave the impression of an exceptionally sound and strong work. I have no doubts that there is a lot of potential in investigating these three avenues, and especially points (1) and (2).

**Weaknesses:**

Unfortunately, despite the impressive claims and indeas from the introduction, I struggle to find in the paper whether these things were actually accomplished or not. I will divide the weaknesses in major and minors:
* Major weakness 1: The claim of "Integration of specialised layer an modules designed to handle noise and distortion more effectively", arguably the most important modification from a technical/research point of view, is never detailed in the paper. Image 1 shows the proposed architecture, but there is no explanation regarding which are the new layers/modules introduced. The fact that they should be "specialised to handle noise and distortion" is an interesting idea but its implementation and functioning need to be explained in detail. How are these components specialised, and why such specialisation should make them handle noise/distortion better? There is no discussion on why particular layers/modules were chosen, and how they influence the final results respect to the original. Note that the Compound Scaling technique cannot be equated to the claim of introducing new layers and moduels specialised as described above. Compound Scaling is a hyperparameter tuning technique which, while surely useful, is not particularly novel or groundbreaking.
* Major weakness 2: The authors describe at length how simulation of rain, droplets, fog, glare, etc. can be applied to augment the dataset and consequently make the model robust against those interferences. However, in the Dataset Description section, the authors only state that they apply "basic data augmentation techniques" (line 353), such as flipping, scaling, brightness and blurring adjustments. This is in direct contrast to the introductory claim of "advanced data augmentation techniques" (line 095). The claim that these augmentations would mimic the effect of rain is clearly not believable and incorrect. Perhaps only blurring could, somehow, mimic reduced visibility thanks to fog or rain, but even so, it depends greatly on the type of blurring applied, which is not described. If any of the important data augmentation techniques (actual simulations of rain, droplets, fog, etc. either via a physics engine or other methods) are applied, it is not evident from the paper.
* Major weakness 3: I disagree with the claim at line 384-385 that the modification led to substantial improvement. The final performance of YOLOv8++ is only very marginally better than the original YOLOv8. No statistical test is performed to check whether the difference shown in the table is significant. I am also unsure about the validity of calculating the average, at the bottom of Table 1, in that way, basically taking the average mAP score of the 5 datasets. Each dataset contains a different number of images, and perhaps a relative weighting should be applied. In addition, and connected to the point 2 above regarding augmentation, as far as I understand the only dataset that natively contains rainy images is DAWN-rainy. Therefore, I think this dataset should be the one used for comparison, as it is a faithful example of real-life rainy conditions, and not a synthetised one. From Table 1, it is clear that performance on DAWN-rainy is equal between the original and proposed method.
* Major weakness 4: There is no mention of the train/test split or other essential components of training, for instance how the Compound Scaling was perormed (and how the corresponding validation dataset is chosen). My suggestion would be to train the model on BDD100K and Cityscapes (with and without augmentations), and use DAWN-rainy as test dataset only. This would give a better estimate of real-world performance of the proposed method.

* Minor weakness 1: The authors spend an enormous amount of space (1 full page, 20% of the paper!) to explain weight pruning, which is an extremely basic and well-known concept. I suggest to summarise weight pruning in no more than two sentences, and use the space for more important and interesting topics, like the explanation of the additional specialised layers/modules.
* Minor weakness 2: The authors base their work on the YOLOv8 model, which seems to be 2 versions behind the current latest YOLO (see line 053). There is no explanation as to why they chose an outdated version. Even assuming that for some reason, the v8 version was the most appropriate, then performance of YOLOv8++ should be compared with that of YOLOv10.
* Minor weakness 3: The claims at lines 400-401 are unsubstantiated, as the authors do not provide a measure of improved robustness or efficiency. The claim of improved accuracy is disputed by the points I raise above. The claims in lines 411-421 are likewise unsubstantiated as no comparison of  efficiency (for instance by meausing training time, number of parameters, or memory footprint) is provided between the original and proposed model.

**Final Rebuttal Confidence:**

4

**Final Rebuttal Justification:**

The author do not implement the ideas they put forward. They claim to integrate "specialized layers and modules that are designed to handle noise and distortions more effectively", but these "layers and modules" are only 1 CNN layer, which is neither specialised not designed. This would have been the one novel, and important, contribution of this paper, but it is not done. The rest of the techniques are interesting, but straightforward application of well established concepts (e.g. weight pruning), which yield results well within expectations. One technique that is perhaps more interesting than the others, namely the generation of rain patterns in the datasets. But no test is performed for other weather conditions for which the authors also claim their method to work, for example fog. All in all, this paper contains good ideas, but most of the claims are not substantiated or verified, and the major point that could be novel is actually not done.

**Justification:**

The ideas of the paper are interesting, and the field of application is both important and actual, but it is not evident from the body of the work that the proposed ideas were actually carried out as presented in the introduction. Several claims are unsubstantiated or wrong. What could be the main technical/research contributions of the work, are not discussed or detailed at a sufficient level. The improvement in the performance appear to be so marginal, that I doubt their statistical relevance. The level of detail is such that the work is not reproducible or verifiable even for an experienced researcher.

---

> ### Author Rebuttal · Authors · 2024-10-24
>
> Comment 1 - The claim of "Integration of specialised layer and modules designed to handle noise and distortion more effectively", arguably the most important modification from a technical/research point of view, is never detailed in the paper. Image 1 shows the proposed architecture, but there is no explanation regarding which are the new layers/modules introduced. The fact that they should be "specialised to handle noise and distortion" is an interesting idea but its implementation and functioning need to be explained in detail. How are these components specialised, and why such specialisation should make them handle noise/distortion better? There is no discussion on why particular layers/modules were chosen, and how they influence the final results respect to the original. Note that the Compound Scaling technique cannot be equated to the claim of introducing new layers and modules specialised as described above. Compound Scaling is a hyperparameter tuning technique which, while surely useful, is not particularly novel or groundbreaking.
>
> Response - In Fig. 1, the first ConvModule is added to the existing YOLOv8 architecture. Adding this to the existing architecture aims to reduce the effect of rain streaks, noise, and distortion created by the data augmentation method. This module would increase the number of architecture parameters, which would be further reduced during the pruning phase. The compound scaling parameters are listed in Table 1 of the revised manuscript.
>
>
> Comment 2 - The authors describe at length how simulation of rain, droplets, fog, glare, etc. can be applied to augment the dataset and consequently make the model robust against those interferences. However, in the Dataset Description section, the authors only state that they apply "basic data augmentation techniques" (line 353), such as flipping, scaling, brightness and blurring adjustments. This is in direct contrast to the introductory claim of "advanced data augmentation techniques" (line 095). The claim that these augmentations would mimic the effect of rain is clearly not believable and incorrect. Perhaps only blurring could, somehow, mimic reduced visibility thanks to fog or rain, but even so, it depends greatly on the type of blurring applied, which is not described. If any of the important data augmentation techniques (actual simulations of rain, droplets, fog, etc. either via a physics engine or other methods) are applied, it is not evident from the paper.
>
> Response - A detailed algorithm for data augmentation is discussed in Algorithm 2.
>
>
> Comment 3 - I disagree with the claim at line 384-385 that the modification led to substantial improvement. The final performance of YOLOv8++ is only very marginally better than the original YOLOv8. No statistical test is performed to check whether the difference shown in the table is significant. I am also unsure about the validity of calculating the average, at the bottom of Table 1, in that way, basically taking the average mAP score of the 5 datasets. Each dataset contains a different number of images, and perhaps a relative weighting should be applied. In addition, and connected to the point 2 above regarding augmentation, as far as I understand the only dataset that natively contains rainy images is DAWN-rainy. Therefore, I think this dataset should be the one used for comparison, as it is a faithful example of real-life rainy conditions, and not a synthetised one. From Table 1, it is clear that performance on DAWN-rainy is equal between the original and proposed method.
>
> Response - With the number of parameters being added now, it can be seen from Tables 2 and 3  that the performance of YOLOv8++  (mAP@50 with 50% pruning - 57.8) is marginally better than YOLOv8 (mAP@50 - 57.6) but with a reduced number of parameters (YOLOv8++ with 50% pruning - 21.897 M params vs YOLOv8 - 43.69 M params). YOLOv8++ achieves the 2x compression ratio in comparison with the YOLOv8 model.
> A basic average is considered over the weighted average as the number of images in datasets is highly varying (BDD100K - 100K, BDD100K-Rainy - 100K, Cityscapes - 5K, Cityscapes-Rainy - 5K and DAWN-Rainy - 200). In this case, a weighted average will be biased towards the 100K dataset, and the contribution of 200 images would be negligible.
>
>
> Comment 4 - There is no mention of the train/test split or other essential components of training, for instance how the Compound Scaling was performed (and how the corresponding validation dataset is chosen). My suggestion would be to train the model on BDD100K and Cityscapes (with and without augmentations), and use DAWN-rainy as test dataset only. This would give a better estimate of real-world performance of the proposed method.
>
> Response - We have utilized three datasets, namely BDD100K, Cityscapes and DAWN-Rainy. BDD100K and Cityscapes datasets are clear weather datasets, while DAWN-Rainy is the real rain dataset. The data augmentations are only applied on BDD100K and Cityscapes datasets, which are called BDD100K-Rainy and Cityscapes-Rainy. The results shown in Table 2 refer to the training and testing (80:20 split ratio) on these five datasets: two real datasets without rain, one real dataset with rain and two rain-augmented datasets.
>
>
> Comment 5 - The authors spend an enormous amount of space (1 full page, 20% of the paper!) to explain weight pruning, which is an extremely basic and well-known concept. I suggest to summarise weight pruning in no more than two sentences, and use the space for more important and interesting topics, like the explanation of the additional specialised layers/modules.
>
> Response – As suggested by the reviewer, the weight pruning is now summarized in Algorithm 1, while the explanation of the additional specialised layers/modules is included in the revised manuscript.
>
>
> Comment 6 - The authors base their work on the YOLOv8 model, which seems to be 2 versions behind the current latest YOLO (see line 053). There is no explanation as to why they chose an outdated version. Even assuming that for some reason, the v8 version was the most appropriate, then performance of YOLOv8++ should be compared with that of YOLOv10.
>
> Response -  In Table 2, the performance of YOLOv8 and YOLOv8++ were compared with the newer versions like YOLO-NAS and YOLOv10, and now the latest version, YOLOv11, is also included in the revised manuscript.
>
>
> Comment 7 - The claims at lines 400-401 are unsubstantiated, as the authors do not provide a measure of improved robustness or efficiency. The claim of improved accuracy is disputed by the points I raise above. The claims in lines 411-421 are likewise unsubstantiated as no comparison of efficiency (for instance by measuring training time, number of parameters, or memory footprint) is provided between the original and proposed model.
>
> Response - The number of parameters of the models is included in Tables 2 and 3 of the revised manuscript.

---

### Official Review · Reviewer_zRDL · 2024-10-07
**YOLOv8++ with Weights Pruning for Road Object Detection in Rainy Environment**

**Confidence:** 4

**Summary:**

The authors present a methodology for image object detection, using the YOLO model, to perform well also under rainy conditions using augmentation techniques, in particular given by horizontal flips, image scaling, brightness adjustments and blurring. The authors also present a pruning method to achieve a sparse neural network for efficiency considerations. The work uses known data sets as benchmarks: BDD100K, Cityscapes016 and DAWN-Rainy. Compared to previous YOLO versions, the methodology, called YOLOv8++, outperforms on average with respect to the data sets. The authors moreover show, for the particular datasets, that the pruning technique does not significantly reduce the performance of the model for up to 50 % pruning.

**Strengths:**

1: The authors describe their work in simple terms, and the purpose is clearly defined, namely the use of augmentation techniques to improve on object detection under rainy conditions. 2: The authors are using several data sets as benchmark which enables a robust comparison with other methods/models. 3: Ablation study of the pruning technique to see the trade-off between degree of pruning and loss in performance.

**Weaknesses:**

1. The methodologies do not present anything particular innovative: The augmentation techniques are well-known. The pruning technique is also well-known. See "Learning both weights and connections for efficient neural networks" by Han et al. published in NIPS'15. This paper should have been cited.
2. The improvement in performance is not particularly large, and whether the method is significantly better than the others could be better inferred if the models where retrained multiple times per data set, and the results where given in mean +- standard deviation.
3. The paper did not include sufficient information on how the augmentation was achieved, and rather the pruning technique took more place. Different augmentation techniques should be considered and compared against each other, for instance in an ablation study.

**Final Rebuttal Confidence:**

4

**Final Rebuttal Justification:**

I consider the revised version as improved with the details in algorithms 1 and 2 as well as Figure 2 in terms of explaining the specific details and making the results reproducible. However, importantly novelty in the paper is still missing. It is still the case that "application of AI"-papers can be highly valuable especially for considering the effect of certain AI methodologies in different scenarios. However, in this case, whether there is a true positive effect of their procedure and how large that would be, is unclear to me. Correctness would be improved by presenting the variation in model performance such as in mean +- std. Ablation study of image augmentation is missing which could tell the significance of each augmentation procedure.

**Justification:**

The assessment is mostly based on the fact that it is not considered as innovative as the techniques presented in the paper do not bring anything new to the table (classic augmentation of images and pruning techniques for neural networks). Moreover, it is unclear whether the model in fact, statistically speaking, significantly outperforms the other models.

---

> ### Author Rebuttal · Authors · 2024-10-24
>
> Comment 1 - The methodologies do not present anything particular innovative: The augmentation techniques are well-known. The pruning technique is also well-known. See "Learning both weights and connections for efficient neural networks" by Han et al. published in NIPS'15. This paper should have been cited.
>
> Response - As suggested by the reviewer, the paper has been cited (reference no. 5) in the revised manuscript.
>
>
> Comment 2 - The improvement in performance is not particularly large, and whether the method is significantly better than the others could be better inferred if the models where retrained multiple times per data set, and the results where given in mean +- standard deviation.
>
> Response - The performance improvement is comparable but with fewer model parameters than the baseline model.
>
>
> Comment 3 - The paper did not include sufficient information on how the augmentation was achieved, and rather the pruning technique took more place. Different augmentation techniques should be considered and compared against each other, for instance in an ablation study.
>
> Response - A detailed algorithm for data augmentation is discussed in Algorithm 2.

---

### Official Review · Reviewer_PW4f · 2024-10-09
**Interesting and Many Branched Work, but Incomplete**

**Confidence:** 4

**Summary:**

The paper attempts to address three areas in the field of object detection:
- Object Detection for Adverse Weather (ODAW)
- Efficiency through Weight Pruning
- Model Changes via Compound Scaling
While the project is ambitious and attempts to tackle important challenges, it appears incomplete and lacks clarity in several critical areas. The combination of multiple partially addressed topics obscures the potential valuable contributions.

**Strengths:**

- Strong Motivation for ODAW: The paper provides a solid rationale for focusing on object detection in adverse weather, highlighting the importance of this challenge.
- Clear Weight Pruning Methodology: The approach to weight pruning is well-described, allowing for replication.
Interesting Findings on Pruning Impact: Results indicate that pruning does not significantly deteriorate the model performance when using the proposed model, which is a noteworthy contribution although limited by the somewhat lacking description of the model. This would be stronger if using the known YOLOv8 for the pruning experiments.

**Weaknesses:**

The authors have attempted several improvements to the object detector model YOLOv8. First, they apply data augmentation to improve performance on data from adverse weather conditions. Second, they apply changes to the model architecture based on the success of efficientnet and finally they apply a weight pruning technique to reduce the size of the proposed model. The experiments indicate that the weight pruning does not severly affect the performance of the model on the datasets BDD100K, CityScapes and DAWN-Rainy. However there are many unsupported claims, the methodological section does not properly cover the experiments and components used and the authors have not chosen experiments that isolate variables and thus reveal the effects they seek to understand and so the learning potential is mostly lost.

**Justification:**

This section is long, but consider it an effort to provide proper feedback aiming for you to gain and share more knowledge from the work you have done:

Overall, the questions being tackled are important and interesting and the methods seem relevant, but the work is incomplete.

The authors attempt to address 3 main points:
1. Object detection for Adverse weather (ODAW).
2. Model Efficiency based on weight pruning
3. Model efficinecy based on compound scaling.

Let us take each main point at a time:
1. Object detection for adverse weather:
	1. Motivation: Good
	3. Method: Lacking. For the paper to be informative about the method used, it should be clear to the reader what has actually been done. The authors quickly refer to how such augmentations are typically done and describe the used method in general terms, but without being specific enough for someone to replicate the method.
	5. Experimental design and evaluation: Lacking. With benchmark datasets and a well established metric, this part should not need a lot of information and it is not too short. However, there are some unclear details here that are very important. The paper states that the target is to apply data augmentations to improve the model for adverse weather conditions without depending on real data with adverse weather. The authors have used datasets with both adverse weather and not. Have the augmentations been applied for training and real adverse weather data for testing? If so, the paper would benefit greatly from making this point more clear as this would then indicate if the augmentation is similar enough to the real thing to allow us to not need the extra real data. If not, I would suggest setting ut experiments to compare the same model trained with and without the augmentation and then test both on the real adverse weather to see if the augmentation helped. Cross-validation would not reveil the useful effects here since the cross-validation would both train and test on the augmented data.
	6. Results: Table 1 relates perhaps more to model comparison than to augmentation in practice. Model comparison we will get back to.
	7. Discussion and reflection: Lacking. it is not obvious what we learn related to the augmentation in Table 1. If it was not meant to be tested it might be better to not put so much weight on it in the rest of the paper.
2. Efficiency with weight pruning (overall, better addressed than the augmentation part)
	1. Motivation: Good.
	3. Method: Good, but with a major problem! It is clear what has been done for the pruning and others could use this work for replication. However, the model part and especially the measure of efficiency is not as clear. To claim anything about efficiency, there should be evaluation based on nr. of parameters or FLOPs required or similar useful measure related to required resources for training or use of the model. Thus it is not clear from the paper if this is a model that has been upscaled in all directions, and then pruned or if it actually downscaled as well. This would be important for the results and should be discussed.
	4. Experimental design and evaluation: Small, but reasonably so. The evaluation metric is clearly stated and the table is relatively self-explanatory although it might be helpful to use the term "compression ratio" instead of "pruning" as this is the term you introduce earlier.
	5. Results: OK (given the experimental section). The presentation of the results is relatively clear although the number of digits should be lower. The high number of digits makes many interesting results in the table less visible while not giving more useful information as the values appear relatively noisy considering the variability of the numbers. It even appears that the deterioration of the model performance is far from monotone.
	6. Discussion and reflection: OK. You show that the pruning does not seem to affect the model performance too much. This is interesting and a good finding to show.
3. Model changes by compound scaling:
	1. Motivation and related work: OK.
	2. Method: Lacking. The components of the scaling is discussed, but not the actual scaling method apart from the final network.
	3. Experimental design and evaluation: Based on the motivation, the target here would be to improve efficiency of the model without a too high penalty on performance. To understand more about the tradeoff between efficiency and performance, the results should again include some measure of size or efficiency of the models. Note that the methodological section does not actually make it clear that the proposed model is more efficient or smaller than the original YOLOv8. The figure might show it for those that know the YOLOv8 intimately, but this should not be assumed. Note that this also influences the takeaways from the weight pruning.
	4. Results: Without a better understanding of the model in terms of size or compute requirements it is difficult to make sense of the numbers and thus gain knowledge from the findings.
	5. Discussion and reflection: Lacking.

The paper addresses important challenges in object detection but falls short due to incomplete methodologies, unclear experimental setups, and insufficient analysis of results. The lack of detailed descriptions prevents replication and understanding of the work's impact. Key metrics and discussions are missing, making it difficult to assess the contributions fully.
Recommendation: Not accepted.

To strengthen the paper, the authors should:

- Provide detailed methodologies for data augmentation in adverse weather and compound scaling.
- Clarify experimental setups, making sure they effectively measure the impact of the proposed methods.
- Include essential metrics such as model size, number of parameters, and computational requirements for efficiency evaluation.
- Reflect on the findings to highlight how each component contributes to new knowledge.
By addressing these issues, the paper could provide more substantial contributions to research.

---

> ### Author Rebuttal · Authors · 2024-10-24
>
> Comment 1 - Object detection for adverse weather Method: Lacking. For the paper to be informative about the method used, it should be clear to the reader what has actually been done. The authors quickly refer to how such augmentations are typically done and describe the used method in general terms, but without being specific enough for someone to replicate the method.
>
> Response - The description regarding the data augmentation can be seen in Algorithm 2 of the revised manuscript.
>
>
> Comment 2 - Experimental design and evaluation: Lacking. With benchmark datasets and a well established metric, this part should not need a lot of information and it is not too short. However, there are some unclear details here that are very important. The paper states that the target is to apply data augmentations to improve the model for adverse weather conditions without depending on real data with adverse weather. The authors have used datasets with both adverse weather and not. Have the augmentations been applied for training and real adverse weather data for testing? If so, the paper would benefit greatly from making this point more clear as this would then indicate if the augmentation is similar enough to the real thing to allow us to not need the extra real data. If not, I would suggest setting up experiments to compare the same model trained with and without the augmentation and then test both on the real adverse weather to see if the augmentation helped. Cross-validation would not reveil the useful effects here since the cross-validation would both train and test on the augmented data.
>
> Response - We have utilized three datasets, namely BDD100K, Cityscapes and DAWN-Rainy. BDD100K and Citiscapes datasets are clear weather datasets while DAWN-Rainy is the real rain dataset. The data augmentations are only applied on BDD100K and Citiscapes datasets and are called BDD100K-Rainy and Cityscapes-Rainy. The results shown in Table 1 refer to the training and testing, with an 80:20 split ratio, on these five datasets: two real datasets without rain, one real dataset with rain and two rain-augmented datasets.
>
>
> Comment 3 - Results: Table 1 relates perhaps more to model comparison than to augmentation in practice.
>
> Response - The paper's primary objective is to detect road objects during rain. That is the reason Table 1 (which is now Table 2) focuses on model comparison. Besides, the data augmentation part is catered via BDD100K-Rainy and Cityscapes-Rainy datasets. The qualitative data augmentation and detection results can be seen in Fig. 2.
>
>
> Comment 4 - Discussion and reflection: Lacking. it is not obvious what we learn related to the augmentation in Table 1. If it was not meant to be tested it might be better to not put so much weight on it in the rest of the paper.
>
> Response – The description regarding the data augmentation can be seen in Algorithm 2 and its qualitative results in Fig. 2 of the revised manuscript.
>
>
> Comment 5 – Efficiency with weight pruning: It is clear what has been done for the pruning and others could use this work for replication. However, the model part and especially the measure of efficiency is not as clear. To claim anything about efficiency, there should be evaluation based on nr. of parameters or FLOPs required or similar useful measure related to required resources for training or use of the model. Thus it is not clear from the paper if this is a model that has been upscaled in all directions, and then pruned or if it actually downscaled as well. This would be important for the results and should be discussed.
>
> Response - The efficiency with weight pruning is measured in terms of the number of parameters and is included in Tables 2 and 3, respectively. Regarding the scaling of the model, the compound scaling parameters comparison can be seen in Table 1.
>
>
> Comment 6 – Experimental design and evaluation: Small, but reasonably so. The evaluation metric is clearly stated and the table is relatively self-explanatory although it might be helpful to use the term "compression ratio" instead of "pruning" as this is the term you introduce earlier.
>
> Response - Besides the number of parameters obtained after pruning, we have included the compression ratio as well in Table 3 of the revised manuscript.
>
>
> Comment 7 – Results: OK (given the experimental section). The presentation of the results is relatively clear although the number of digits should be lower. The high number of digits makes many interesting results in the table less visible while not giving more useful information as the values appear relatively noisy considering the variability of the numbers. It even appears that the deterioration of the model performance is far from monotone.
>
> Response - The representations are made in percentage for better visibility. The possible reasons for the deterioration of the model performance are far from monotone, could be:
> During the initial stages of pruning, removing these less important weights may have little to no effect on performance, or performance might even improve due to overfitting reduction. This means the model may perform just as well or even better after some pruning, before eventually starting to degrade.
> The model’s layers and connections are highly interdependent. Removing weights from certain layers may not impact the performance much while pruning from other sensitive layers can cause a significant drop in accuracy. This layer sensitivity contributes to the non-monotonic pattern.
> After weight pruning, models are fine-tuned or retrained to help them recover from performance loss. In some cases, this fine-tuning can lead to a temporary recovery or improvement in performance after pruning, causing fluctuations in the performance graph.
>
>
> Comment 8 –Model changes by compound scaling: Method: Lacking. The components of the scaling is discussed, but not the actual scaling method apart from the final network.
>
> Response – The compound scaling parameters are added in Table 1 of the revised manuscript.
>
>
> Comment 9 – Experimental design and evaluation: Based on the motivation, the target here would be to improve efficiency of the model without a too high penalty on performance. To understand more about the tradeoff between efficiency and performance, the results should again include some measure of size or efficiency of the models. Note that the methodological section does not actually make it clear that the proposed model is more efficient or smaller than the original YOLOv8. The figure might show it for those that know the YOLOv8 intimately, but this should not be assumed. Note that this also influences the takeaways from the weight pruning.
>
> Response - The models' efficiency regarding the number of parameters is included in Tables 2 and 3 of the revised manuscript.
>
>
> Comment 10 – Results: Without a better understanding of the model in terms of size or compute requirements it is difficult to make sense of the numbers and thus gain knowledge from the findings.
>
> Response - The number of parameters of the models is included in Tables 2 and 3 of the revised manuscript.
>
>
> Comment 11 – To strengthen the paper, the authors should:
> Provide detailed methodologies for data augmentation in adverse weather and compound scaling.
> Clarify experimental setups, making sure they effectively measure the impact of the proposed methods.
> Include essential metrics such as model size, number of parameters, and computational requirements for efficiency evaluation.
> Reflect on the findings to highlight how each component contributes to new knowledge. By addressing these issues, the paper could provide more substantial contributions to research.
>
> Response - As suggested by the reviewer, the required changes have been made in the revised manuscript.

---

### Meta-Review · Area_Chair_YF74 · 2024-11-01

**Recommendation:** Reject
**Confidence:** 5

**Metareview:**

This paper deals with improving YOLOv8 object detector to perform road object detection under rainy conditions.

Several positive aspects (motivation and importance of the problem, clarity and simplicity of the model, use of public datasets) were identified by the reviewers.

However, more numerous negative aspects were also raised. While the authors provide a systematic rebuttal to reviewers' comments and their additional details were judged valuable, there are still some major concerns that remain after the rebuttal, among which are the lack of novelty, and the lack of detail/justification of the methodology and in the experiments. For instance, adding a convolutional layer is not original nor specialized to handle noise or distortion. Weight pruning has been extensively studied. Comparison with rainy-compliant models is limited. Data augmentations seem basic with respect to the objective of simulating rainy conditions.

Based on these weaknesses, all reviewers suggest rejection.

**Suggested Changes To The Recommendation:**

2: I'm certain of the recommendation.  It should not be changed

---

### Decision · Program_Chairs · 2024-11-06

Reject